# [^99m^Tc]Tc-iFAP/SPECT Tumor Stroma Imaging: Acquisition and Analysis of Clinical Images in Six Different Cancer Entities

**DOI:** 10.3390/ph15060729

**Published:** 2022-06-09

**Authors:** Paola Vallejo-Armenta, Guillermina Ferro-Flores, Clara Santos-Cuevas, Francisco Osvaldo García-Pérez, Pamela Casanova-Triviño, Bayron Sandoval-Bonilla, Blanca Ocampo-García, Erika Azorín-Vega, Myrna Luna-Gutiérrez

**Affiliations:** 1Department of Radioactive Materials, Instituto Nacional de Investigaciones Nucleares, Ocoyoacac 52750, Mexico; paovallejoarmenta@gmail.com (P.V.-A.); clara.cuevas@inin.gob.mx (C.S.-C.); blanca.ocampo@inin.gob.mx (B.O.-G.); 2Department of Nuclear Medicine, Instituto Nacional de Cancerología, Tlalpan, Mexico City 14080, Mexico; fosvaldogarcia@gmail.com (F.O.G.-P.); zafily3@gmail.com (P.C.-T.); 3Department of Neurosurgery, Hospital de Especialidades del Centro Médico Nacional Siglo XXI, IMSS, Cuauhtémoc, Mexico City 06720, Mexico; bayronsandoval@gmail.com

**Keywords:** FAP, ^99m^Tc-FAP inhibitor, ^99m^Tc-labeled iFAP, tumor microenvironment, SPECT

## Abstract

Fibroblast activation protein (FAP) is highly expressed on the cancer-associated fibroblasts (CAF) of the tumor stroma. Recently, we reported the preclinical evaluation and clinical biokinetics of a novel ^99m^Tc-labeled FAP inhibitor radioligand ([^99m^Tc]Tc-iFAP). This research aimed to evaluate [^99m^Tc]Tc-iFAP for the tumor stroma imaging of six different cancerous entities and analyze them from the perspective of stromal heterogeneity. [^99m^Tc]Tc-iFAP was prepared from freeze-dried kits with a radiochemical purity of 98 ± 1%. The study included thirty-two patients diagnosed with glioma (*n* = 5); adrenal cortex neuroendocrine tumor (*n* = 1); and breast (*n* = 21), lung (*n* = 2), colorectal (*n* = 1) and cervical (*n* = 3) cancer. Patients with glioma had been evaluated with a previous cranial MRI scan and the rest of the patients had been involved in a [^18^F]FDG PET/CT study. All oncological diagnoses were corroborated histopathologically. The patients underwent SPECT/CT brain imaging (glioma) or thoracoabdominal imaging 1 h after [^99m^Tc]Tc-iFAP administration (i.v., 735 ± 63 MBq). The total lesions (*n* = 111) were divided into three categories: primary tumors (PT), lymph node metastases (LNm), and distant metastases (Dm). [^99m^Tc]Tc-iFAP brain imaging was positive in four high-grade WHO III–IV gliomas and negative in one treatment-naive low-grade glioma. Both [^99m^Tc]Tc-iFAP and [^18^F]FDG detected 26 (100%) PT, although the number of positive LNm and Dm was significantly higher with [^18^F]FDG [82 (96%)], in comparison to [^99m^Tc]Tc-iFAP imaging (35 (41%)). Peritoneal carcinomatosis lesions in a patient with recurrent colorectal cancer were only visualized with [^99m^Tc]Tc-iFAP. In patients with breast cancer, a significant positive correlation was demonstrated among [^99m^Tc]Tc-iFAP uptake values (Bq/cm^3^) of PT and the molecular subtype, being higher for subtypes HER2+ and Luminal B HER2-enriched. Four different CAF subpopulations have previously been described for LNm of breast cancer (from CAF-S1 to CAF-S4). The only subpopulation that expresses FAP is CAF-S1, which is preferentially detected in aggressive subtypes (HER2 and triple-negative), confirming that FAP+ is a marker for poor disease prognosis. The results of this pilot clinical research show that [^99m^Tc]Tc-iFAP SPECT imaging is a promising tool in the prognostic assessment of some solid tumors, particularly breast cancer.

## 1. Introduction

Tumors are pathological complexes composed of tumor cells and the tumor stroma or tumor microenvironment (TME), which consists of cellular and acellular components, such as cancer-associated fibroblasts (CAFs), endothelial cells, adipocytes, mesenchymal stem cells (MSC), macrophages, blood vessels, pericytes, and extracellular matrices (ECM) [1,2]. In fact, CAFs induce a cancer phenotype and are responsible for the production of proteolytic enzymes, growth factors, and extracellular matrix components [2]. CAFs contribute up to 90% of the macroscopic tumor mass, provide mechanical support to tumor cells and control their survival, metastasis, proliferation, and resistance to therapies. CAFs can have different origins, including adipose mesenchymal stem cells, resident tissue fibroblasts and epithelial/endothelial cells, and adipocytes and pericytes that transdifferentiate to mesenchymal cells; therefore, they represent a heterogeneous cell population within the TME [1].

Fibroblast activation protein (FAP) is a membrane-anchored peptidase expressed by CAFs at the stromal level of various tumor entities and contributes to progression and a worse prognosis. FAP degrades denatured collagens and participates in tumor growth via a non-enzymatic mechanism [1,2,3].

Diagnostic FAP inhibitor radiotracers under clinical evaluation use ^18^F and ^68^Ga linked to quinolinoyl-cyanopyrrolidine [3,4,5,6] and cyclo-[benzene(trimethanethiol-DOTA)-Met-Pro-Pro-Thr-Glu-Phe-Met] (FAPI-2286) structures [7], which are radiotracers for PET (positron emission tomography), and only one work has reported ^99m^Tc, also linked to quinolinoyl-cyanopyrrolidine for SPECT (single-photon emission computed tomography) imaging [3]. 

Internationally, the amount of equipment available for molecular imaging studies is predominantly higher for gamma cameras (SPECT modality), and they represent more than 70% of the total. For SPECT images, the most-employed radionuclide is ^99m^Tc. Therefore, the need for target-specific radiopharmaceuticals labeled with ^99m^Tc is increasing within the field of oncology. Our group previously reported [^99m^Tc]Tc-((R)-1-((6-hydrazinylnicotinoyl)-D-alanyl)pyrrolidin-2-yl) boronic acid ([^99m^Tc]Tc-iFAP) as a new SPECT radioligand capable of specifically detecting FAP expressed by CAFs located in the cancer stroma and, to our knowledge, the first ligand based on ^99m^Tc-labeled boron-Pro derivatives [8]. Furthermore, the [^99m^Tc]Tc-iFAP biokinetic–dosimetric evaluation in healthy volunteers and three cancer patients diagnosed with breast, lung, and cervical cancer showed favorable biokinetics and uptake in primary tumor lesions and lymph node metastases, achieving high-quality and high-contrast molecular images [9].

This research aimed to evaluate [^99m^Tc]Tc-iFAP for the tumor stroma imaging of six different cancerous entities and analyze them from the perspective of stromal heterogeneity.

## 2. Results

No adverse events related to the diagnostic use of [^99m^Tc]Tc-iFAP were observed. 

Table 1 shows the general characteristics of patients included in the [^99m^Tc]Tc-iFAP imaging evaluation. A detailed cancer staging of patients is shown in Table A1 (Appendix A). Patient imaging results were categorized into two groups. Patients with gliomas (*n* = 5), with which SPECT and SPECT/MR images were acquired, were identified as the first group. The second group involves all cases (*n* = 27) of breast, lung, colon, NET, renal cortex, and cervical cancer, in which SPECT/CT and PET/CT images were obtained.

[^99m^Tc]Tc-iFAP SPECT brain imaging was positive in four high-grade WHO III–IV gliomas (T/Bc range 6.3–13.9) (Table 2) and negative in one treatment-naive low-grade glioma (Figure 1). [^99m^Tc]Tc-iFAP imaging resolution and contrast were good enough for the high-grade glioma, which could allow the performing of non-invasive diagnoses to differentiate between low- and high-grade gliomas based on their distinct FAP expression [10]. 

For all cancer cases, a total of 111 lesions were evaluated, which were classified as primary tumors (PT)(*n* = 26), lymph node metastases (LNm) (*n* = 61), and distant metastases (Dm) (*n* = 24) (Table 3). 

All primary tumors were detected with both [^99m^Tc]Tc-iFAP SPECT/CT and [^18^F]FDG PET/CT (Figure 2 and Figure 3), which did not occur with LNm and Dm lesions (Figure 4 and Figure 5). That is, [^99m^Tc]Tc-iFAP SPECT/CT detected PT (100%), LNm (51%), and Dm (17%) in contrast to [^18^F]FDG PET/CT, which detected PT (100%), LNm (100%), and Dm (88%) (Table 3). 

The non-detection of LNm and Dm with [^99m^Tc]Tc-iFAP could be attributed to the lower spatial resolution of the SPECT technique in 61% of the lesions (size < 8 mm), including those not detected in NT and lung cancer, but not in 39% of the lesions with dimensions greater than 8 mm and associated to breast cancer. Additionally, none of the Dm lesions detected by [^18^F]FDG in patients with triple negative and luminal B HER2+ molecular subtypes at the bone, liver, and lung exhibited [^99m^Tc]Tc-iFAP uptake.

These results are expected since FAP expression decreases once the cells succeed to invade. FAP is a protein that promotes metastasis; therefore, once the micrometastasis is established in a distant site from the PT, it loses its FAP expression. The signaling mediated by FAP/integrins/PI3K has a negative effect on IGF2 expression (associated with increased glucose uptake). This fact probably explains why as FAP uptake in Dm decreases, FDG uptake increases [11]. 

As a unique feature of [^99m^Tc]Tc-iFAP images, a very low background was achieved as previously reported (Figure 3) [9]. 

[^99m^Tc]Tc-iFAP uptake was considerably lower regarding [^18^F]FDG in patients with cervical cancer and neuroendocrine tumor (NET) of the adrenal cortex, which agrees with their relatively low FAP expression in comparison to lung and breast cancer (Figure 2) [12,13]. 

In general, the Dm lesions detected with [^18^F]FDG did not show [^99m^Tc]Tc-iFAP uptake, except for peritoneal carcinomatosis lesions in recurrent colorectal cancer, which only showed [^99m^Tc]Tc-iFAP uptake, but not [^18^F]FDG uptake (Figure 5). CAFs are abundant in mesothelial metastases, and, through the mesothelial-mesenchymal transformation mechanism, it is likely that in carcinomatosis there is a greater transdifferentiation of mesenchymal cells towards CAFs FAP+ [14].

When comparing the values obtained from the average tumor-to-background ratios of the different background sites [T/Bm (tumor/mediastinum), T/Bl (tumor/liver) and T/Bp (tumor/psoas muscle)] for all lesions, the highest values were T/Bp for both imaging methods (Figure 6). Although no statistically significant difference was found, the values of the T/Bp ratios were higher with [^18^F]FDG than with [^99m^Tc]Tc-iFAP (Figure 6). In the T/B data, the same trend is observed in terms of a higher uptake of [^99m^Tc]Tc-iFAP in the primary tumors compared to that obtained in the lymph node and distant metastases (Figure 6).

In the case of breast cancer, [^99m^Tc]Tc-iFAP showed a significant positive correlation between the T/Bp value of the primary tumors and the molecular subtype (Pearson correlation coefficient: *r* = 0.8085), where HER2+ and Luminal B HER2+ enriched subtypes showed the highest T/Bp ratios (Figure 7). The [^99m^Tc]Tc-iFAP uptake in HER2+ could be associated to the Erb2-mediated phosphorylation of Tyr654 of β-catenin, which promotes the activation of Wnt signaling pathways and the consequent promotion of the tumor invasive capacity (FAP expression) through a very common mechanism in breast cancer, the epithelial–mesenchymal transition (EMT) process, induced by the microenvironment, which infers the gain of invasive capacity and the arrest of the cell cycle, while, at the signaling level, it implies the repression of E-cadherin expression through snail/slug [14,15]. 

In LN metastases, a decrease in T/Bp ratio was observed and there was no significant correlation among the molecular subtypes (Pearson correlation coefficient: *r* = 0.4027) (Figure 7). 

## 3. Discussion

The expression of FAP is an indication that the cell is expressing an invasive phenotype associated with an intense process of differentiation, typical of the first stages of carcinogenesis [16]. During this phase, there is an intense activation of signaling pathways aimed at promoting the differentiation of cell precursors towards the activated fibroblast phenotype. As the tumor evolves, the stroma changes genetically and epigenetically to generate the appropriate niche for its stage. Cellular plasticity allows cells to adapt to their microenvironment through reprogramming processes (phenotypic and genotypic modifications) for tumor progression. RNAs produce epigenetic modifications that alter transcription, activating stem cell transformation and EMT processes (including FAP expression), which are essential for invasion to occur [16]. FAP is overexpressed by CAFs from various tumor entities, making it a promising biomarker and target for many medical interventions. CAF subpopulations (from CAF-S1 to CAF-S4) are classified depending on the expression of six markers: integrin b1/CD29, α-SMA (alpha-smooth muscle actin), PDGFR-β (platelet-derived growth factor receptor β), fibroblast activation protein (FAP), CAV1 (caveolin 1), and S100-A4/FSP1 (fibroblast-specific protein 1). The only subpopulation that expresses FAP is CAF-S1 (CAF-S1 FAP+) [17].

In their study, Kratochwil et al. demonstrated the elevated and selective uptake of ^68^Ga-FAPI-04 in the stroma of multiple tumors, including breast, lung, colorectal, and NET cancer [13]. However, this research demonstrated the tumor stroma imaging with [^99m^Tc]Tc-iFAP as the first SPECT radioligand based on a boron-Pro derivative [8].

The results showed that the detection of primary tumor lesions with [^99m^Tc]Tc-iFAP is consistent when compared with [^18^F]FDG. However, when detecting LNm and Dm, the superiority of [^18^F]FDG is clear. This fact can be attributed to the lower spatial resolution of the SPECT technique in 61% of the lesions (size < 8 mm), but not in 39% of the tumors with dimensions greater than 8 mm and associated to breast cancer. Thus, our findings are discussed from the perspective of tumor stroma heterogeneity in lesions with enough size to be detected by SPECT. 

As mentioned, the dynamics of differentiation in the tumor microenvironment are attributed to genetic and non-genetic changes in tumor cells, the composition of the extracellular matrix, cell–cell interactions, and cell heterogeneity [16]. Based on this, it is likely that the increased uptake of [^99m^Tc]Tc-iFAP by primary tumors due to the presence of increased amounts of CAF-S1 FAP+ indicates an active EMT process, which is known to happen in the early phases of carcinogenesis, through which the dissemination of cells from the primary mass to distant sites is promoted [15]. EMT involves the regulation of both intercellular adhesions by decreasing E-cadherin and increasing N-cadherin, as well as substrate adhesions through integrin mediated primarily by TGF-B, β-catenin, and the Wnt signaling pathway.

On the other hand, hypoxic and hypoglycemic tumor stroma synergistically promotes the EMT phenotype in carcinomas. Thus, tumors where GLUT1 expression is commonly increased will also have an inability to express an (invasive) EMT phenotype [14]. Accordingly, it is likely that the lack of uptake of [^99m^Tc]Tc-iFAP in LNm and Dm is related to the fact that in this type of lesion there is an increase in the expression of GLUT1 receptors that leads to an increase in glucose metabolism, which produces a rise in the uptake of [^18^F]FDG and, at the same time, inhibits EMT (including FAP expression). 

The neoplasm with the largest number of patients in this study was breast cancer, which showed a significant positive correlation in PT between the T/Bp value and the molecular subtypes, with the highest T/Bp ratios for the HER2+ and Luminal B subtypes HER2+. The T/Bp values in HER2+ breast cancer showed a significant decrease in the LN metastases regarding PT (Figure 7), which may be due to crosstalking (cross-regulation), which occurs between integrins and EGFR receptors, such as HER2 [18,19]. Additionally, it is known that the Wnt signaling pathway promotes the proliferation and invasion of breast cancer cells in a HER2-dependent manner [20]. It was recently confirmed that the expression of HER2 in the cell membrane is heterogeneous and that the accumulation of HER2 occurs in regions where adhesion to the extracellular matrix is dynamic [21]. Therefore, HER2 expression decreases in regions where focal adhesions are concentrated, and the relative local decrease in HER2 expression in LNm, compared to PT, is probably related to the metastatic process. 

As a relevant point, it is noted that the presence of CAFs in the tumor stroma of breast cancer is associated with resistance to immunotherapy [22], since the elements secreted by CAFs derived from HER2+ tumors regulate resistance to treatment in a paracrine way. Thus, the decrease in [^99m^Tc]Tc-iFAP uptake by LNm could indicate greater sensitivity to trastuzumab treatment.

Highly-relevant data have been reported on axillary LNm in breast cancer: (1) the stroma represents around 25–30% of the invaded areas (regardless of subtype); (2) the predominant CAF subpopulations are CAF-S1 and CAF-S4 (the latter being the most abundant); (3) CAFs enrichments are different in LNm compared to PT; (4) the secretion of CXCL12β by CAF-S1 and the expression of CXCR4 in cancer cells is involved in the initiation of EMT and in the distant metastatic process, particularly in lung and bone; and (5) the global stromal content in LNm provides a prognostic stratification of breast cancer patients and, therefore, the CAF-S1/CAF-S4 abundance status exhibits a prognostic value, since both present pro-invasive properties with different modes of action; however, CAF-S4 is known to have a greater impact on distant metastatic spread (Dm), particularly on the liver [17,23].

CAF-S1 FAP+ promotes an immunosuppressive environment by secreting CXCL12β, promoting the presence of CD4+CD25+ T cells, increasing T cell survival, and promoting the cell differentiation into CD25+FOXP3+ cells. The ability of regulatory T cells (Tregs) to inhibit the proliferation of effector T cells is also enhanced by CAF-S1. CAF-S4 is highly contractile and induces cancer cell invasion in three dimensions through Notch signaling. CAF-S1 FAP+ is preferentially detected in aggressive subtypes (HER2+ and triple negative), confirming that FAP+ is a poor prognostic marker [17,23].

Summarizing, the relatively low performance of [^99m^Tc]Tc-iFAP in detecting LNm and Dm may be related to the molecular biology of cancer and the proportion of the enrichment of CAF-S1 FAP+, which is not the most abundant in metastatic lesions (LN or distant) (Figure 8). Even when the FAP expression is associated with a phenotype that tends to transmigration and proliferation, attention must be placed on the fact that its expression is temporary and that it depends largely on the tumor microenvironment dynamics; thus, when the characteristics of the tumor stroma are modified, FAP expression and cancer prognosis can change.

Our results differ from the work of Kömek et al. [24], where they showed that PET/CT [^68^Ga]Ga-FAPI-04 is superior to [^18^F]FDG in the detection of primary mammary lesions and metastases (ganglionic and visceral) in twenty patients with breast cancer, both in primary and recurrent lesions, although the average size of the evaluated LNm was 10 mm [24]. On the other hand, Backhaus et al. [25] evaluated the use of PET/MRI with the ligand [^68^Ga]Ga-FAPI-46 PET/CT in 19 women with breast cancer with evidence of high uptake in the primary lesions (mean diameter of 26 mm) and LNm (average diameter of 21 mm). Our results probably vary from the previous research carried out due to the heterogeneity of the sample with respect to the molecular subtypes of breast cancer, the image acquisition time, and the different image acquisition method (SPECT/CT vs. PET/CT vs. PET/MRI). The dynamic behavior of FAP is firmly associated with its functions in the progression phase during cancer evolution (tissue remodeling, extracellular matrix degradation, the promotion of tumor proliferation, and immunomodulation) [15,16,17,18,19,20,21,22,23], which deserves to be used as a tool for the detection of the heterogeneity of the tumor stroma in the different stages of cancer through molecular imaging with specific radiotracers, such as [^99m^Tc]Tc-iFAP. Therefore, additional clinical studies must be performed, including the results of the ex vivo FAP expression in tumors (immunohistochemical evaluation) to be correlated with the uptake of FAP inhibitory radiotracers. 

Today, CAFs is receiving considerable attention in the field of cancer biology. Targeted CAF therapy can potentially inhibit metastasis and cancer progression by reducing immunosuppression and remodeling the tumor microenvironment. Therapeutic targeting of FAP has been described in different modalities, such as vaccines, oncolytic viruses, and nanoparticles [26]. In preclinical studies, CAF-S1 FAP+ has shown to cause resistance to anti-PD-L1 immunotherapy and reduce antitumor immunity. CAFs from breast, ovarian, lung, pancreas, and colon cancer have shown expression of PD-L1 and/or PD-L2; particularly the CAF-S1 FAP+ subset. Additionally, the CAF-S1 FAP+ subpopulation is an important source of CXCL12 secretion, which plays a crucial role in resistance to anti-PD-1 and anti-CTLA-4 immunotherapies in pancreatic, ovarian, and breast cancer [27,28].

Taking into account the deleterious effect of metastases on the survival of breast cancer patients, our data could heighten the interest in evaluating the abundance of the CAF-S1 FAP+ subpopulation, in vivo, in a non-invasive manner, by means of [^99m^Tc]Tc-iFAP SPECT in axillary LNm during the initial clinical approach (staging) to determine the prognosis and the benefit of therapies, such as anti-FAP, anti-TGFβ, anti-CXCR4, and/or anti-PD-L1 immunotherapy, in combination with standard therapies (Figure 8). More prospective research is needed to enrich the information obtained so far and we believe that future research can be focused on the function of FAP ligands in different molecular and histological subtypes of breast cancer, as well as their potential in detecting relapse of the disease, in the evaluation of the response to therapy and the prognosis of the patient.

Peritoneal carcinomatosis is a complication of various malignant tumors and is generally associated with a poor prognosis. The superiority of uptake by [^99m^Tc]Tc-iFAP in peritoneal carcinomatosis, due to recurrent colon cancer observed in the patient included in this study, agrees with the findings previously described, demonstrating a greater sensitivity of [^68^Ga]Ga-FAPI-04 for the detection of peritoneal carcinomatosis in patients with various types of cancer [29].

The findings observed in patients with glioma coincide with the data previously reported by Röhrich et al., where they showed little or no uptake of ^68^Ga-FAPI-02 and FAPI-04 in low-grade WHO II gliomas and high uptake in gliomas of high WHO III-IV grade, regardless of HDI status [10]; therefore, its usefulness could lie mainly in the differentiation of tumor recurrence versus post-treatment changes and in surgical and/or radiotherapy planning, for which more prospective studies are needed in this regard.

## 4. Materials and Methods

### 4.1. Reagents

An iFAP (boron-Pro ligand) lyophilized kit for ^99m^Tc labeling was obtained from the National Institute of Nuclear Research (ININ, Ocoyoacac, Mexico) with GMP certification [8]. [^99m^Tc]TcO_4_Na was eluted from a generator (^99^Mo/^99m^Tc GETEC, ININ, Ocoyoacac, Mexico). Other reagents were received from Millipore Co. (Burlington, MA, USA). 

### 4.2. [^99m^Tc]Tc-iFAP Preparation

After the reconstitution of the iFAP lyophilized kit with a [^99m^Tc]TcO_4_Na/0.2 M phosphate buffer (1:1 *v*/*v*, 2 mL, 740 MBq) solution and incubation in a block heater (92 °C, 15 min), the [^99m^Tc]Tc-iFAP radioligand was obtained with a radiochemical purity (R.P.) greater than 98% (HPLC, Discovery C18 column, 5 µm particle size, I.D. of 0.46 cm, length of 25 cm; Supelco, Millipore, Burling-ton, MA, USA; coupled to a UV–Vis detector and a radiometric detector), applying the following linear gradient: a flow rate of 1 mL/min, 0.1% TFA/water (A) (from 100 to 50%, over 10 min, maintained for 10 min, 30% over 5 min, and returned to 100% over 5 min) and 0.1% TFA/acetonitrile (B). As previously reported, the lyophilized formulation contains the HYNIC-iFAP (((R)-1-((6-hydrazinylnicotinoyl)-*D*-alanyl)pyrrolidin-2-yl)boronic) ligand with a specific alignment to the corresponding regions of the FAP binding site [8], stannous chloride as a reducing agent, as well as ethylenediaminediacetic acid (EDDA) to complete the coordination sphere of the [Tc(V)]HYNIC core (Figure 9).

The chemical characterization of the iFAP ligand included analysis by mass spectrometry (UPLC-mass), ^1^H–NMR, UV–Vis and FT-IR. Radiochemical characterization included reversed-phase radio-HPLC and ITLC-SG (instant thin layer chromatography-silica gel) with the following mobile phases: 2-butanone, 0.1 M sodium citrate, and ammonium acetate-methanol (1:1 *v*/*v*), as reported in detail previously [8].

### 4.3. Patients

Thirty-two patients (mean ± SD age, 50.8 ± 16.7 years; 28 women and 4 men) with different types of cancer (breast cancer (*n* = 21), lung cancer (*n* = 2), adrenal cortex NETs (*n* = 1), colorectal cancer (*n* = 1), cervical cancer (*n* = 3) and gliomas (*n* = 5)) were included.

The patients were divided into two groups as follows: Group 1 (*n* = 5 gliomas) and Group 2 (*n* = 27 breast, lung, colon, renal cortex NET, and cervical cancer). The characteristics of the patients are shown in Table 1 and with a detailed clinical description in Appendix A (Table A1). All oncological diagnoses were determined histopathologically (Table 2).

The patients underwent SPECT/CT 1–3 *h* (with an average of 2 h) after the intravenous application of [^99m^Tc]Tc-iFAP (735 ± 63.5MBq). In Group 1, only of the brain region, and in Group 2, the thoracoabdominal region. The tumor/background ratio is optimal for diagnostic images from 30 min post-injection [9]. However, it was decided 2 h after radiotracer administration to improve the contrast of the images (lesions vs. background). The acquisition protocol and the post-injection waiting time were the same for all types of cancer evaluated. However, in patients with cervical cancer or pelvic etiology, immediate image acquisition was performed post-micturition to reduce the artifact of radiotracer accumulation in the urine.

All patients in Group 1 had previous cranial MRI (6 ± 1 days interval) and patients in Group 2 had previous [^18^F]FDG PET/CT studies carried out (11 ± 12.6 days interval).

This research was performed in the Department of Nuclear Medicine of the National Cancer Institute (INCan), Mexico. The patients signed an informed consent declaration, and the protocol was approved by the institutional Nuclear Medicine Ethics Committee. 

### 4.4. Image Acquisition

[^99m^Tc]Tc-iFAP SPECT/CT images were acquired with a dual-head gamma camera (SPECT/CT, Symbia TruePoint, Siemens, Malvern, PA, USA), with low-energy, high-resolution collimators; parameters: window at 140 keV, matrix size of 128 × 128, with dispersion correction, 90 images of 8 s, rotation of 360 degrees. For the attenuation correction map, the low-dose CT parameters were obtained. A Butterworth filter (cutoff: 0.5, 5th order) and an iterative method (8 iterations /4 subsets) were used for the reconstruction of the raw data.

SPECT/CT images were acquired 2 h after the intravenous administration of [^99m^Tc]Tc-iFAP (735 ± 63.5 MBq). The anatomical region studied in Group 1 was only the brain and in Group 2 it was thoracoabdominal. Activity in regions of interest was quantified, via 3D imaging, as Bq/cm^3^.

All patients in Group 2 had undergone a prior PET/CT (Excel 20) scan (Siemens Medical Solutions), performed at 1 h after [^18^F]FDG administration (CT: slice thickness of 5 mm, 180 mAs and 120 kVp). Whole-body scans were obtained in 3D mode from the vertex to mid-thighs (2–3 min per bed position). PET images were reconstructed using a two-dimensional expectation algorithm of ordered subsets.

### 4.5. Image Analysis

Images obtained with [^99m^Tc]Tc-iFAP and [^18^F]FDG were examined on a Siemens VG60 multimodal workstation. Visual and semi-quantitative analyses were performed by two physicians with more than 9 years of experience in nuclear medicine and molecular imaging (workstation with processing software for volumetric analysis).

Visual analysis was performed on both groups of patients. Uptake was compared with the morphology of the corresponding lesion using CT and/or MRI, depending on the patient group. The detected lesions were divided into three categories for study: primary tumor (PT), lymph node metastases (LNm), and distant metastases (Dm). The semiquantitative analysis of lesion uptake was obtained by calculating the tumor-to-background ratio (T/B) with spherical volumes of interest (VOIs) to homogenize the data obtained with both radiopharmaceuticals. Additionally, in Group 2, the concordance of uptake between both radiotracers was compared by quantifying the number of lesions (PT, LNm, and Dm).

### 4.6. Tumor Tissue Samples

All patients underwent a biopsy of the primary tumor lesion. Histopathology was used to determine the existence of viable tumor tissue and to verify the diagnosis. The histopathological reports were interpreted by a certified and experienced pathologist.

### 4.7. Statistical Analysis

The Pearson correlation coefficient was calculated between the T/Bp [^99m^Tc]Tc-iFAP values and the molecular subtypes of the patients with breast cancer; a value of *p* < 0.05 was considered statistically significant.

## 5. Conclusions

The results of this pilot study show that SPECT imaging with [^99m^Tc]Tc-iFAP is a promising and potentially useful tool in the evaluation of the tumor microenvironment of multiple solid neoplastic entities. Within the different types of cancer that we included, we observed a potential panorama in the prognostic evaluation of recently diagnosed breast cancer, as well as its probable diagnostic superiority in peritoneal carcinomatosis in recurrent colon cancer. [^18^F]FDG was superior to [^99m^Tc]Tc-iFAP in the detection of LNm and Dm. However, with the analyses carried out, we can establish that the role of [^99m^Tc]Tc-iFAP is not to displace metabolic molecular imaging, but rather that it serves as a complement for an adequate prognostic evaluation.

Further prospective [^99m^Tc]Tc-iFAP clinical studies are needed to define the clinical impact of the non-invasive in vivo detection of FAP in newly diagnosed breast cancer patients and its implication in determining candidates for immunotherapy and target therapy combined with conventional therapies.

## Figures and Tables

**Figure 1 pharmaceuticals-15-00729-f001:**
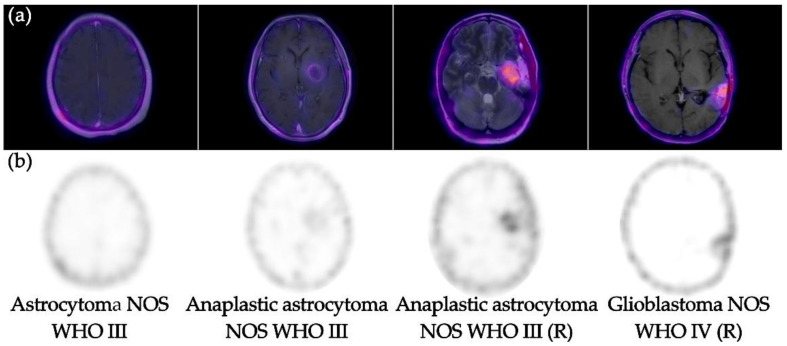
(**a**) [^99m^Tc]Tc-iFAP SPECT coregistered to MR images (SPECT/MRI) and (**b**) [^99m^Tc]Tc-iFAP SPECT. Note the adequate visualization of [^99m^Tc]Tc-iFAP uptake in high-grade gliomas (WHO III-IV)-treatment-naive and recurrent (R). However, low-grade glioma (WHO II) did not show uptake.

**Figure 2 pharmaceuticals-15-00729-f002:**
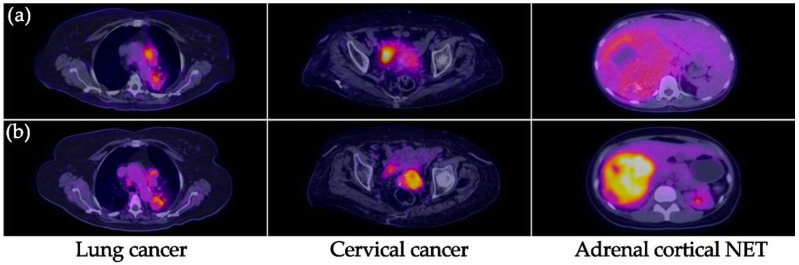
(**a**) [^99m^Tc]Tc-iFAP SPECT/CT and (**b**) [^18^F]FDG PET/CT images of the primary tumors of three different types of cancers. All primary lesions show concordant uptake between both molecular imaging methods. [^99m^Tc]Tc-iFAP uptake was considerably lower regarding [^18^F]FDG in patients with cervical cancer and neuroendocrine tumor (NET) of the adrenal cortex.

**Figure 3 pharmaceuticals-15-00729-f003:**
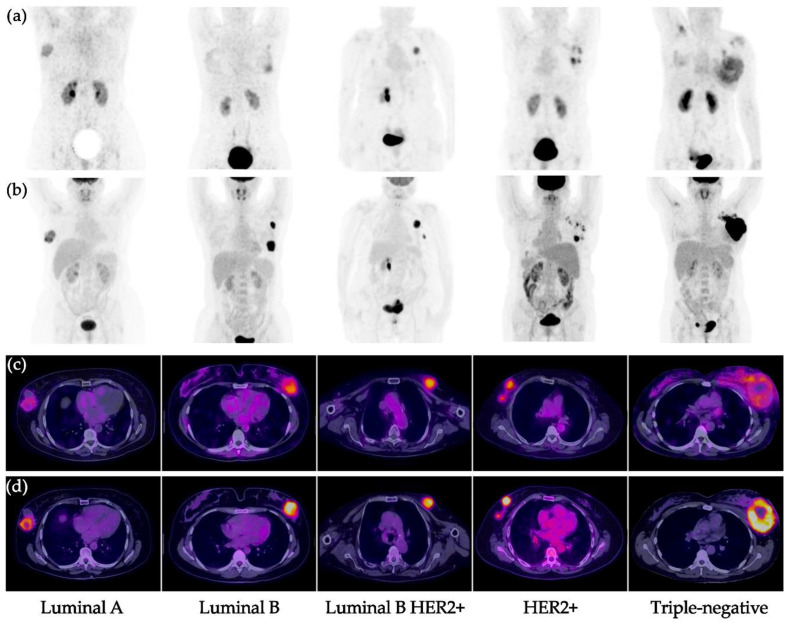
Primary breast cancer tumors. (**a**) [^99m^Tc]Tc-iFAP MIP, (**b**) [^18^F]FDG MIP, (**c**) [^99m^Tc]Tc-iFAP SPECT/CT, and (**d**) [^18^F]FDG PET/CT. All primary lesions show concordant uptake between both molecular imaging methods. [^99m^Tc]Tc-iFAP uptake is decreased in pure hormonal molecular subtypes (Luminal A and B) and elevated in subtypes with HER2+ expression (Luminal B HER2+ and pure HER2+). The triple negative subtype shows moderate and heterogeneous uptake. MIP: maximum intensity projection.

**Figure 4 pharmaceuticals-15-00729-f004:**
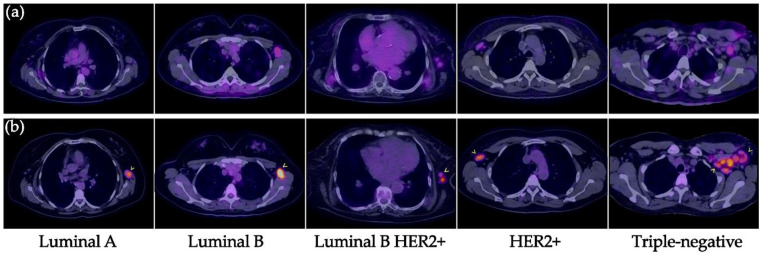
Lymph node metastases in breast cancer. (**a**) [^99m^Tc]Tc-iFAP SPECT/T and (**b**) [^18^F]FDG PET/CT. All malignant-appearing axillary lymphadenopathies are hypermetabolic; however, most of them (arrowheads) exhibit reduced or absent [^99m^Tc]Tc-iFAP uptake in all molecular subtypes of breast cancer (lesion sizes >8 mm).

**Figure 5 pharmaceuticals-15-00729-f005:**
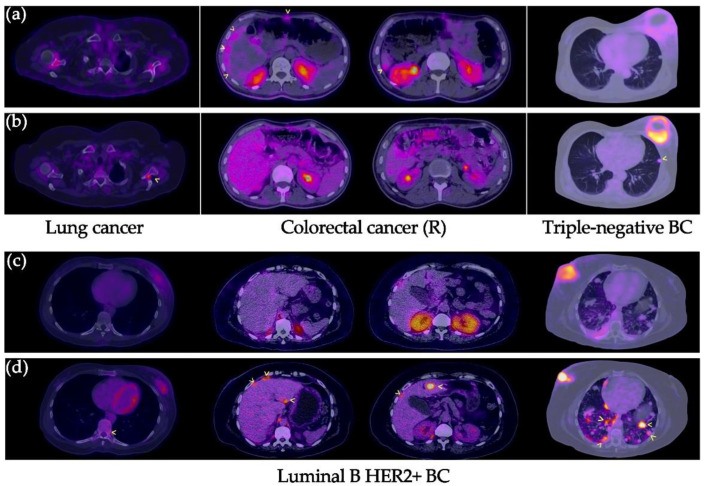
Distant metastases in various types of cancers. (**a**,**c**) [^99m^Tc]Tc-iFAP SPECT/CT and (**b**,**d**) [^18^F]FDG PET/CT. All distant metastatic lesions are hypermetabolic; however, most of them (arrowhead) exhibit decreased or no uptake of [^99m^Tc]Tc-iFAP. In the case of the patient with recurrent colon cancer, areas of diffuse [^99m^Tc]Tc-iFAP uptake were observed in liver subcapsular implants and in the anterior abdominal wall, which were not detected with [^18^F]FDG. BC: breast cancer. R: recurrence.

**Figure 6 pharmaceuticals-15-00729-f006:**
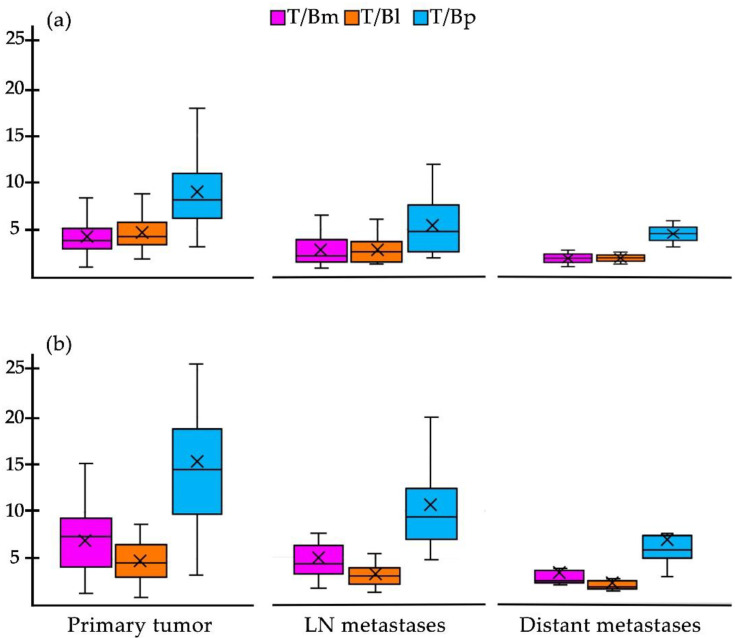
Box plot of the target-to-background ratios of (**a**) [^99m^Tc]Tc-iFAP and (**b**) [^18^F]F-FDG in all primary tumors, lymph node metastases, and distant metastases (except gliomas).The T/Bp ratio is higher in all categories of both radiotracers, particularly in the primary tumor. T/Bm (tumor/mediastinum), T/Bl (tumor/liver), and T/Bp (tumor/psoas muscle). ×= mode.

**Figure 7 pharmaceuticals-15-00729-f007:**
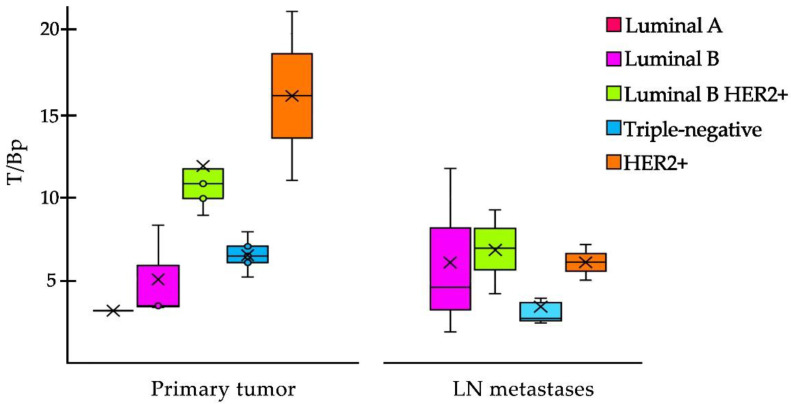
Box plot of the T/Bp ratio of [^99m^Tc]Tc-iFAP in primary tumors and LN metastases of breast cancer. In primary tumors, the T/Bp (tumor/psoas muscle) ratio is higher in HER2+ and Luminal B HER2+ molecular subtypes. In LN metastases, a decrease in T/Bp is observed and there is no significant difference among the molecular subtypes (Pearson correlation coefficient: *r* = 0.4027). ×= mode and ° = outliers.

**Figure 8 pharmaceuticals-15-00729-f008:**
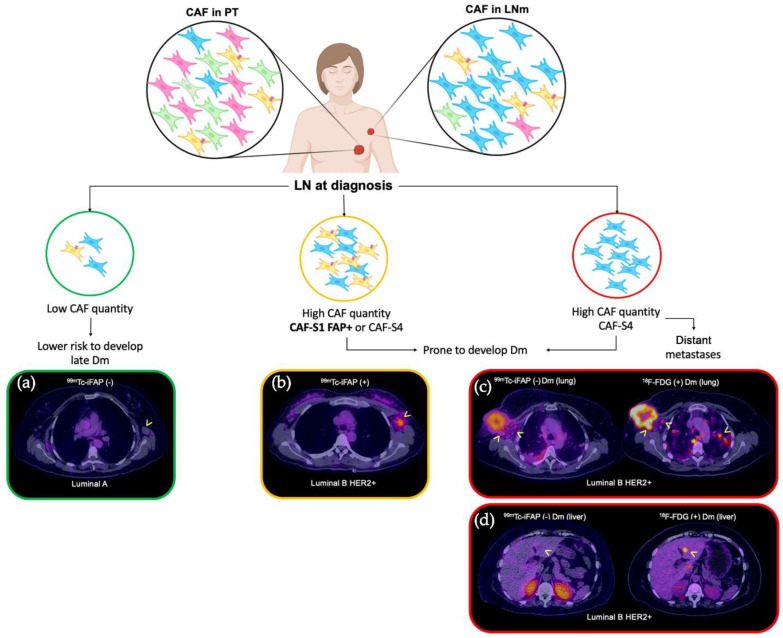
CAF subpopulations as prognostic markers in breast cancer (in diagnosis). Four CAF subpopulations have been reported in the lymph node metastases of breast cancer (CAF-S1 to CAF-S4). The most relevant and predominant are CAF-S1 FAP+ and CAF-S4 FAP-. Pelon et al. [23] established a model of clinical application to the knowledge generated from the different subpopulations, in such a way that a prognostic impact is proposed according to the predominance of CAFs as follows: if at the time of diagnosis the patient exhibits low content of CAF-S1 FAP+ in LNm, they present a low risk of late Dm ((**a**) [^99m^Tc]Tc-iFAP SPECT/CT(-), no uptake in left axillary adenopathy of Luminal A breast cancer); on the other hand, if high levels of CAF-S1 FAP+ are demonstrated in LNm, the risk of distant metastasis increases ((**b**) [^99m^Tc]Tc-iFAP SPECT/CT(+), uptake in left axillary adenopathy of Luminal B HER2+ breast cancer). Finally, in distant metastatic lesions, only CAF-S4 FAP- is expressed [(**c**) [^99m^Tc]Tc-iFAP SPECT/CT(-) in lung Dm and extremely low uptake in some right axillary lymph nodes that exhibit hypermetabolism with [^18^F]FDG; likewise, (**d**) multiple lung and liver metastases did not exhibit uptake of [^99m^Tc]Tc-iFAP]. BC: breast cancer, LNm: lymph node metastasis, Dm: distant metastasis.

**Figure 9 pharmaceuticals-15-00729-f009:**
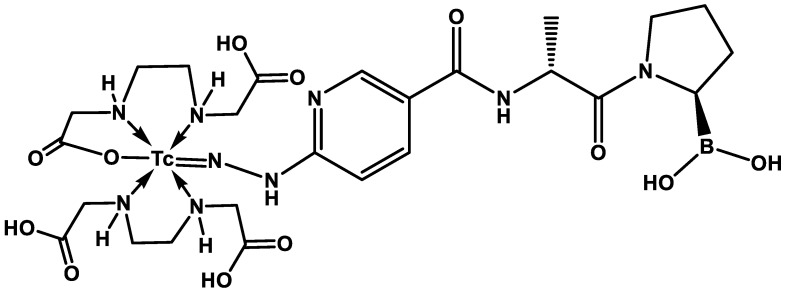
The proposed [Tc(V)]EDDA/HYNIC core structure in the [^99m^Tc]Tc-iFAP radioligand. The advantage that HYNIC-iFAP presents with respect to previously reported quinolinoyl-cyanopyrrolidine-based FAP inhibitors is the possibility of obtaining [^99m^Tc]Tc(V)-EDDA/HYNIC stable cores from instant freeze-dried kit formulations.

**Table 1 pharmaceuticals-15-00729-t001:** General characteristics of the patients included in the [^99m^Tc]Tc-iFAP imaging study.

Characteristics	Number
No. patient	32
**Age** (**years**)	50.8 ± 16.7
**Gender** (**%**)	
Female	28 (88%)
Male	4 (12%)
**Diagnosis**	**Cases** (**%**)
**Breast cancer**	21 (66%)
Ductal carcinoma, Luminal A	2
Ductal carcinoma, Luminal B	3
Ductal carcinoma, Luminal B HER2+	5
Ductal carcinoma, HER2+	2
Ductal carcinoma, Triple negative	9
**Lung cancer**	2 (6%)
NSCLC adenocarcinoma	
**Cervical cancer**	3 (9%)
Squamous cell carcinoma	
**Glioma**	5 (16%)
Astrocytoma NOS (WHO II)	1
Anaplastic astrocytoma NOS (WHO III)	3
Glioblastoma NOS (WHO IV)	1
**Colorectal cancer**	1 (3%)
Adenocarcinoma	1
**Adrenal cortical neuroendocrine tumor**	1 (3%)
Poorly differentiated, Ki67 30%	1
**Clinical setting** (**%**)	
Initial staging	27 (84%)
Restaging	5 (15%)

**Table 2 pharmaceuticals-15-00729-t002:** Tumor-to-contralateral tissue background ratio (T/Bc) of [^99m^Tc]Tc-iFAP in patients with high-grade WHO III–IV gliomas.

Diagnosis	Status Brain SPECT	T/Bc
Astrocytoma NOS (WHO II)	Negative	NA
Anaplastic astrocytoma NOS (WHO III) (*n* = 2)	Positive	6.3 and 7.8
Anaplastic astrocytoma NOS restaging (WHO III)	Positive	15.4
Glioblastoma NOS (WHO IV)	Positive	13.9

**Table 3 pharmaceuticals-15-00729-t003:** Number of lesions detected with [^99m^Tc]Tc-iFAP and [^18^F]FDG in all patients except gliomas.

	Primary Tumor	Lymph Node Metastases	Distant Metastases	Total
**All lesions (N)**	**26**	**61**	**24**	**111**
[^99m^Tc]Tc-iFAP	26 (100%)	31 (51%)	4 (17%)	61 (55%)
[^18^F]FDG	26 (100%)	61 (100%)	21 (88%)	108 (97%)
	**Diagnosis**	**Lymph node metastases**	**Distant metastases**	
[^99m^Tc]Tc-iFAP		*n* = 31 (51%)	*n* = 4 (17%)	
	Lung cancer NSCLC	3	0	
	Triple-negative BC	10	0	
	Luminal A	0	0	
	Luminal B HER2+ BC	7	2	
	Luminal B BC	4	0	
	HER2+ BC	5	0	
	Cervical cancer	2	0	
	Colorectal cancer	0	3	
	Adrenal cortical NT	0	1	
[^18^F]FDG		*n* = 61 (100%)	*n* = 21 (88%)	
	Lung cancer NSCLC	3	1	
	Triple-negative BC	25	1	
	Luminal A BC	5	0	
	Luminal B HER2+ BC	8	16	
	Luminal B BC	10	1	
	HER2+ BC	7	0	
	Cervical cancer	2	0	
	Colorectal cancer	0	0	
	Adrenal cortical NT	1	2	

BC: breast cancer; NSCLC: non-small cell lung cancer; NT: neuroendocrine tumor.

## Data Availability

Data are contained within the article.

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
