# Peer review of "[99mTc]Tc-iFAP/SPECT Tumor Stroma Imaging: Acquisition and Analysis of Clinical Images in Six Different Cancer Entities"

_pharmaceuticals, 2022, doi:10.3390/ph15060729_

Round 1

Reviewer 1 Report

Authors have written a research article on “[99mTc]Tc-iFAP/SPECT Tumor Stroma Imaging: Acquisition and Analysis of Clinical Images in Six Different Cancer Entities”. The manuscript has been written well. Authors have explained every part very nicely. The manuscript can be accepted for publication after addressing the following comment.

Minor comments

1.     Authors have mentioned “After reconstitution of the iFAP lyophilized kit with a [99mTc]TcO4Na/0.2 M phos-phate buffer (1:1 v/v, 2 mL, 740 MBq) solution and incubation in a block heater (92°C, 15 min), the [99mTc]Tc-iFAP radioligand was obtained” in [99mTc]Tc-iFAP preparation method. What type of reaction/mechanism or bond formation between iFAP and [99mTc]Tc involved in the preparation method? Please mention.

2.      In HPLC analysis of [99mTc]Tc-iFA, which detector was used?

3.     Other than HPLC,does authors have performed any test for conforming synthesis of the [99mTc]Tc-iFAP radioligand. Such as TLC/NMR/Mass spectrometry.

Author Response

Reviewer #1:

Authors have written a research article on “[99mTc]Tc-iFAP/SPECT Tumor Stroma Imaging: Acquisition and Analysis of Clinical Images in Six Different Cancer Entities”. The manuscript has been written well. Authors have explained every part very nicely. The manuscript can be accepted for publication after addressing the following comment.

Minor comments

  1. Authors have mentioned “After reconstitution of the iFAP lyophilized kit with a [99mTc]TcO4Na/0.2 M phosphate buffer (1:1 v/v, 2 mL, 740 MBq) solution and incubation in a block heater (92°C, 15 min), the [99mTc]Tc-iFAP radioligand was obtained” in [99mTc]Tc-iFAP preparation method. What type of reaction/mechanism or bond formation between iFAP and [99mTc]Tc involved in the preparation method? Please mention.

ANSWER: In agreement with the reviewer, on page 13, line 364, the following paragraph was added:

“As previously reported [8], the lyophilized formulation contains the HYNIC-iFAP ligand, stannous chloride as a reducing agent, as well as ethylenediaminediacetic acid (EDDA) to complete the coordination sphere of the [Tc(V)]HYNIC core (Figure 9) [8].”

  1. Trujillo-Benítez, D.; Luna-Gutiérrez, M.; Ferro-Flores, G.; Ocampo-García, B.; Santos-Cuevas, C.; Bravo-Villegas, G.; Morales-Ávila, E.; Cruz-Nova, P.; Díaz-Nieto, L.; García-Quiroz, J.; et al. Design, Synthesis and Preclinical Assessment of 99mTc-iFAP for In Vivo Fibroblast Activation Protein (FAP) Imaging. Molecules 2022, 27, 264. https://doi.org/10.3390/molecules27010264.
  2. In HPLC analysis of [99mTc]Tc-iFA, which detector was used?

ANSWER: In agreement with the reviewer, on page 13, line 361, the following paragraph was added:

 “HPLC…. coupled to a UV-Vis detector and a radiometric detector”

  1. Other than HPLC,does authors have performed any test for conforming synthesis of the [99mTc]Tc-iFAP radioligand. Such as TLC/NMR/Mass spectrometry.

ANSWER: In agreement with the reviewer, on page 13, line 370, the following paragraph was added:

 “The chemical characterization of the iFAP ligand included analysis by mass spectrometry (UPLC-mass), 1H–NMR, UV-Vis and FT-IR [8]. Radiochemical characterization included reversed-phase radio-HPLC and ITLC-SG (instant thin layer chromatography-silica gel) with the following mobile phases: 2-butanone, 0.1 M sodium citrate, and ammonium acetate-methanol (1:1 v/v ), as reported in detail previously [8].”

  1. Trujillo-Benítez, D.; Luna-Gutiérrez, M.; Ferro-Flores, G.; Ocampo-García, B.; Santos-Cuevas, C.; Bravo-Villegas, G.; Morales-Ávila, E.; Cruz-Nova, P.; Díaz-Nieto, L.; García-Quiroz, J.; et al. Design, Synthesis and Preclinical Assessment of 99mTc-iFAP for In Vivo Fibroblast Activation Protein (FAP) Imaging. Molecules 2022, 27, 264. https://doi.org/10.3390/molecules27010264.

Reviewer 2 Report

The paper entitled “[99mTc]Tc-iFAP/SPECT Tumor Stroma Imaging: Acquisition and Analysis of Clinical Images in Six Different Cancer Entities” by Vallejo-Armenta et al. presents the clinical evaluation of  [99mTc]Tc-iFAP, a novel 99mTc-based FAP imaging agent in different tumor entities.

Overall, the paper is written well, concisely and clear. The results and discussion indicating the use of this imaging agent as a complement tracer for creating an adequate prognosis is  convincing, the paper should be published in Pharmaceutics after the following minor revisions:

 Since [99mTc]Tc-iFAP is structurally different to the more common quinolinoyl-cyanopyrrolidine based FAP inhibitors, please add a figure showing the structure of [99mTc]Tc-iFAP and explain shortly if there are advantages of using the boron-Pro lead motif.

Although given correctly in the legend of the figures, please add the names [99mTc]Tc-iFAP and [18F]FDG e.g. at the left side of the matrix in figures 1-6 so that it gets clear from the first view which row belongs to which tracer.

Tumor tissue samples were obtained form all patient’s primary tumor lesions. Have FAP expression levels been analyzed in these samples? If yes, do FAP protein levels correlate with uptake levels of [99mTc]Tc-iFAP?

Furthermore:  Line 342… please change “in patients in with” to ‘in patients with” as well as line 363… please change to “5 min) and 0.1%”.

Author Response

Reviewer #2:

The paper entitled “[99mTc]Tc-iFAP/SPECT Tumor Stroma Imaging: Acquisition and Analysis of Clinical Images in Six Different Cancer Entities” by Vallejo-Armenta et al. presents the clinical evaluation of  [99mTc]Tc-iFAP, a novel 99mTc-based FAP imaging agent in different tumor entities.

Overall, the paper is written well, concisely and clear. The results and discussion indicating the use of this imaging agent as a complement tracer for creating an adequate prognosis is  convincing, the paper should be published in Pharmaceutics after the following minor revisions:

 Since [99mTc]Tc-iFAP is structurally different to the more common quinolinoyl-cyanopyrrolidine based FAP inhibitors, please add a figure showing the structure of [99mTc]Tc-iFAP and explain shortly if there are advantages of using the boron-Pro lead motif.

ANSWER: In agreement with the reviewer, on page 13, a figure of the proposed [99mTc]Tc-iFAP chemical structure was added (Figure 9) including the following figure caption:

“Figure 9. Proposed [Tc(V)]EDDA/HYNIC core structure in the [99mTc]Tc-iFAP radioligand [8]. The advantage that HYNIC-iFAP presents with respect to previously reported quinolinoyl-cyanopyrrolidine based FAP inhibitors is the possibility of obtaining [99mTc]Tc(V)-EDDA/HYNIC stable cores from instant freeze-dried kit formulations.”

Although given correctly in the legend of the figures, please add the names [99mTc]Tc-iFAP and [18F]FDG e.g. at the left side of the matrix in figures 1-6 so that it gets clear from the first view which row belongs to which tracer.

ANSWER: In agreement with the MDPI editorial, we are following the Microsoft Word template, which clearly establishes the following in the guidelines for figure captions:

If there are multiple panels, they should be listed as: (a) Description of what is contained in the first panel; (b) Description of what is contained in the second panel.”

Furthermore, the addition of the names [99mTc]Tc-iFAP and [18F]FDG at the left side of the matrix in figures 1-6, would reduce the area and appreciation of molecular images.

Tumor tissue samples were obtained form all patient’s primary tumor lesions. Have FAP expression levels been analyzed in these samples? If yes, do FAP protein levels correlate with uptake levels of [99mTc]Tc-iFAP?

ANSWER: Histopathological studies were conducted in all cases, but the evaluation of FAP expression via Immunohistopathological studies was not performed.  Therefore, on page 11, line 296, the following paragraph is written:

“Probably our results vary from the previous research carried out due to heterogeneity of the sample with respect to the molecular subtypes of breast cancer, the image acquisition time, and the different image acquisition method (SPECT/CT vs PET/CT vs PET/MRI). Nevertheless, additional clinical studies must be performed, including results of the ex-vivo FAP expression in tumors (immunohistochemical evaluation) to be correlated with the uptake of FAP inhibitory radiotracers. “

Furthermore:  Line 342… please change “in patients in with” to ‘“in patients with” as well as line 363… please change to “5 min) and 0.1%”.

ANSWER: They were corrected.

Reviewer 3 Report

The paper is very interesting and adds further knowledge in the field of FAPi imaging. Technetium labelling can appear as a step backward in the era of PET, but the developing technology CZT detectors renders this approach up-to-date.

        The only aspect worthwhile to be elucidated is related to the scan time after injection. You previously reported a washout from the target and a dynamic binding to that. For which reason the 2 hours interval has been chosen? for the best target-to-background ratio or other reasons? this interval can be applied to any tumor type?

Table 3.  third last row, mistype error:  cervicall

Author Response

Reviewer #3:

The paper is very interesting and adds further knowledge in the field of FAPi imaging. Technetium labelling can appear as a step backward in the era of PET, but the developing technology CZT detectors renders this approach up-to-date.

        The only aspect worthwhile to be elucidated is related to the scan time after injection. You previously reported a washout from the target and a dynamic binding to that. For which reason the 2 hours interval has been chosen? for the best target-to-background ratio or other reasons? this interval can be applied to any tumor type?

ANSWER: : In agreement with the reviewer, on page 14, line 391, the following paragraph was added:

“The tumor/background ratio is optimal for diagnostic images from 30 min post-injection [9]. However, it was decided 2 h after radiotracer administration to improve the contrast of the images (lesions vs. background). The acquisition protocol and the post-injection waiting time were the same for all types of cancer evaluated. However, in patients with cervical cancer or pelvic etiology, immediate image acquisition was performed post-micturition to reduce the artifact of radiotracer accumulation in the urine.”

  1. Coria-Domínguez, L.; Vallejo-Armenta, P.; Luna-Gutiérrez, M.; Ocampo-García, B.; Gibbens-Bandala B.; García-Pérez, F.; Ramírez-Nava, G.; Santos-Cuevas, C.; Ferro-Flores, G. [99mTc]Tc-iFAP Radioligand for SPECT/CT Imaging of the Tumor Microenvironment: Kinetics, Radiation Dosimetry, and Imaging in Patients. Pharmaceuticals 2022, 15, 590. https://doi.org/10.3390/ph15050590.

 Table 3.  third last row, mistype error:  cervicall

ANSWER: It was corrected